# Acoustically-Activated Liposomal Nanocarriers to Mitigate the Side Effects of Conventional Chemotherapy with a Focus on Emulsion-Liposomes

**DOI:** 10.3390/pharmaceutics15020421

**Published:** 2023-01-27

**Authors:** Mah Noor Zafar, Waad H. Abuwatfa, Ghaleb A. Husseini

**Affiliations:** 1Biomedical Engineering Program, American University of Sharjah, Sharjah P.O. Box 26666, United Arab Emirates; 2Department of Chemical and Biological Engineering, American University of Sharjah, Sharjah P.O. Box 26666, United Arab Emirates; 3Materials Science and Engineering Program, College of Arts and Sciences, American University of Sharjah, Sharjah P.O. Box 26666, United Arab Emirates

**Keywords:** drug-delivery, eLiposomes, emulsion liposomes, ultrasound

## Abstract

To improve currently available cancer treatments, nanomaterials are employed as smart drug delivery vehicles that can be engineered to locally target cancer cells and respond to stimuli. Nanocarriers can entrap chemotherapeutic drugs and deliver them to the diseased site, reducing the side effects associated with the systemic administration of conventional anticancer drugs. Upon accumulation in the tumor cells, the nanocarriers need to be potentiated to release their therapeutic cargo. Stimulation can be through endogenous or exogenous modalities, such as temperature, electromagnetic irradiation, ultrasound (US), pH, or enzymes. This review discusses the acoustic stimulation of different sonosensitive liposomal formulations. Emulsion liposomes, or eLiposomes, are liposomes encapsulating phase-changing nanoemulsion droplets, which promote acoustic droplet vaporization (ADV) upon sonication. This gives eLiposomes the advantage of delivering the encapsulated drug at low intensities and short exposure times relative to liposomes. Other formulations integrating microbubbles and nanobubbles are also discussed.

## 1. Introduction

Recent statistics show that cancer is the second leading cause of death globally, with 18,094,716 million diagnosed cancer cases. It is responsible for one in six deaths, up to approximately 10 million deaths in 2020 [1]. This multifactorial disease has become a global burden with high cancer morbidity and mortality rates. The heterogeneity of cancer requires extensive research to develop effective treatment methods to reduce the detriment to patients’ lives. Currently, available cancer treatment strategies include chemotherapy [2], radiotherapy [3,4], surgery [5], hormonal therapy, targeted therapy (e.g., immunotherapy) [6], or a combination of these methods [7,8,9].

Chemotherapy is the most used cancer treatment, especially for advanced-stage malignancies, i.e., metastasis, where other treatment methods like surgery and radiation cannot be employed. It uses anticancer drugs to intervene with the cancer cell cycle [10]. Anticancer drugs are highly toxic, have short half-lives in vivo, have poor biodistribution, and have low bioavailability. Moreover, these highly toxic drugs lack selectivity, distribute in the entire body, and kill both cancerous and healthy cells, inducing severe temporary or permanent side effects among long-term cancer survivors [11,12,13]. Some of the long-term side effects include pain, hair loss, nausea, vomiting, pulmonary toxicity, neuropathy, and cardiotoxicity [13,14,15,16,17]. Cancer cells possess self-renewal properties and evade drug-induced cytotoxicity, enabling them to develop resistance against a wide range of anticancer drugs in a multidrug resistance (MDR) phenomenon, rendering the chemotherapeutic drug ineffective [18]. Thus, conventional cancer treatments are not optimal and suffer from limitations, including the lack of specificity, systemic toxicity, harmful side effects, and tumor recurrence. To combat these limitations, drugs must be targeted and temporally released at the tumor site using safe trigger mechanisms [19].

Research aims to develop less toxic alternative treatment platforms, including smart nanotechnology for anticancer drug delivery, that can potentially address the challenges posed by cancer cells’ uncontrolled proliferation, metastasis, and MDR [20,21]. Smart nanotechnology enables the encapsulation and targeted delivery of anticancer drugs. Chemotherapeutic drugs act on healthy and tumor cells, thus reducing the treatment selectivity towards diseased cells. Doxorubicin (DOX) is a commonly used anthracycline that is usually administered to patients via intravenous injections, either as a continuous infusion or a single dose [22]. Although it is an effective antitumor agent, its side effects are most evident in cells exhibiting high division rates, such as hair follicles and the gastrointestinal tract lining; thus, hair loss, digestive tract ulcerations, vomiting, nausea, and diarrhea are all common complications/side effects [23]. Moreover, it has been known to induce cardiotoxicity by the upregulation of apoptosis receptors in cardiomyocytes [23]. The therapeutic index of chemotherapeutic drugs, defined as the ratio of the toxic dosage to the therapeutic dosage, is close to one. Therefore, maximum tumor cell killing while protecting healthy cells cannot be achieved due to the side effects limiting the chemotherapeutic drug dosage [23]. A study investigating the toxicity profiles of free chemotherapeutics versus chemotherapeutics delivered in nanovehicles reported safer and more effective chemotherapy using nanoparticles, in vitro and in vivo [24]. Nanoparticles were loaded with 30% of the standard dose of cisplatin and administered for 4 weeks and were found to eradicate as many cancer cells as the higher dosage of cisplatin. Another in vivo study on rats employed chemotherapy using inactive cisplatin (a prodrug that activates upon release). The drug was injected as a free drug and via nanoparticles [25,26]. Figure 1 illustrates the concept of nanoparticle-based chemotherapy delivery to solid tumors. After encapsulating the drug into the nanocarriers, they can be steered to localize at the diseased site. Upon accumulation, internal or external triggers can stimulate the release of the loaded drugs. These nanoplatforms allow for targeted delivery of the anticancer agents as well as controlled dosing.

The United States Food and Drug Administration (FDA) has approved the use of some nano-drug formulations to treat various diseases, including cancer. Current FDA-approved nanocarriers include 56% lipid-based, 38% protein-based, and 6% metal-based formulations. Some of the lipid-based nano-drugs, namely liposomal formulations approved by the FDA and/or EMA from 1995 to May 2022 for the treatment of cancer, include Doxil, DaunoXome, Myocet, Mepact, Marqibo, Onyvide, and Vyxeos (Table 1) [27,28]. Table 1 provides a summary of the liposomal nanopharmaceuticals approved by the FDA and/or EMA for cancer treatment. These nanopharmaceuticals showed clinical success due to their biocompatibility, selective toxicity, stability, biodegradability, and prolonged blood circulation times [29]. Moreover, high drug loading capacity and bioavailability at the tumor site by crossing physiological barriers leads to efficient, targeted drug delivery and pared down off-target effects [30,31].

## 2. Liposomal-Based Smart Drug Delivery

Smart drug delivery systems (SDDSs) are nanoplatforms with essential characteristics and bio-functions that make them optimal for encapsulating and shielding the proper dosage of chemotherapeutic agents from healthy tissues, while remotely delivering drugs to targeted tumor sites under controlled release conditions. SDDSs overcome shortcomings of traditional treatment approaches, as they are designed to deliver appropriate dosages to specific anatomical locations, combat systemic side effects, prolong the circulation time of the drug and make it more bioavailable [18]. They incorporate nanoparticles (NPs) as drug-delivery vehicles, usually ranging in size from 1 nm to 800 nm. The synthesis routes of these NPs vary and are generally divided into chemical and biological ways, where the latter are preferred as they are safer and innocuous [19].

The morphologies of the NPs play a substantial role in the success of the SDDS. For instance, size and shape properties determine essential parameters such as circulation time, the efficiency of targeting, and internalization by the cells. The optimal size range for NPs is between 100 and 200 nm; as NPs exceeding 7 μm are expelled by the lungs, less than 6 nm are filtered by the kidneys, and between 0.1 and 7 μm are recognized by the reticuloendothelial system (RES) and phagocytized [23]. Moreover, regularly shaped NPs, as in spherical or cylindrical NPs, exhibit better performance than irregularly shaped ones, as the former are promptly internalized by the cells and move easily through the endothelial lining of the blood vessels. Likewise, surface functionalities such as hydrophobicity, charge, and targeting moieties can potentially alter the performance of the NPs [19,23].

Among the most researched NPs are lipid-based nanocarriers, including FDA-approved liposomes, which have shown significant success in the pharmaceutical industry. These nanocarriers are nanoscale spherical vesicles primarily compromising one or more phospholipid bilayers similar to the structure of the cell membranes [41,42]. They can be synthesized using natural substances or sometimes synthetic surfactants that entrap molecules with low molecular weights, drugs, imaging agents, vaccines, plasmid DNA, hormones, antibodies, etc. [43,44]. Phospholipids are the main structural components of liposomes, where the most commonly used are soy phosphatidylcholine (soy PC) [39], egg phosphatidylcholine (egg PC) [40], phosphatidylcholine (PC), and milk fat phospholipids [41]. Phospholipids occur abundantly in nature; however, their synthetic derivatives also exist, such as 1,2-Dimyristoyl-sn-glycero-3-phosphocholine (DMPC) and 1,2-Dipalmitoyl-sn-glycero-3-phosphocholine (DPPC) as derivatives of PC, and 1,2-Dimyristoyl-sn-glycero-3-phosphoethanolamine (DMPE) as a derivative of PE, etc. [42].

Liposomes are versatile drug delivery vehicles as they have adequate stability, can prolong the drug’s bioavailability and protect its rapid clearance from the body, are easily tunable, and exhibit controlled release profiles that lead to remarkable therapeutic effects with minimum unwanted side effects [43,44]. Due to the resemblance of cell membranes, liposomes are also referred to as artificial cells and were previously called Banghasomes; Alec D. Bangham discovered them in the 1960s when Bangham and colleagues observed the swelling of lipid vesicles upon hydration. They were later named liposomes originating from the Greek words “lipos” and “soma,” meaning fat and body, respectively [45].

Liposomes are amphiphilic in nature and consist of a hydrophilic head (water-loving) and a hydrophobic/lipophilic (water-repelling) tail [46,47]. In an aqueous solution, lipid molecules self-assemble due to hydrophobic interactions, forming bilayer spheres with the hydrophobic tails directed inwards, acting as a permeability barrier, and hydrophilic heads facing outwards. The bilayer structure enables liposomes to encapsulate lipid-soluble and aqueous-soluble drugs simultaneously [48]. The hydrophilic, amphipathic, and lipophilic molecules can be entrapped within the inner aqueous core or within the phospholipid bilayer, which forges the effectiveness of drug loading ahead [49,50,51]. Encapsulating active biomedical drugs within liposomes enables control over targeting cancer cells as it stabilizes the encapsulated chemotherapeutic and can spatially release the payload in a controlled manner, which will not only enhance the efficacy and therapeutic index of drugs but also eliminate or minimize systemic toxicity [52,53].

They can be classified based on lamellarity, size, and preparation method. However, size and lamellarity significantly affect the blood circulation time and drug encapsulation efficiency, whereas the preparation method determines the liposome type. Liposomes can be classified as unilamellar vesicles (UV) with a single phospholipid bilayer membrane or multilamellar vesicles (MLV) with several bilayer membranes resembling an onion-like structure. ULVs can be further classified as small unilamellar vesicles (SUV, diameter of 20–100 nm), large unilamellar vesicles (LUV, diameter 100 nm–1 µm), giant unilamellar vesicles (GUV diameter > 1 µm, [24], and multivesicular vesicles (MVV, 1.6–10.5 µm) that demonstrate a honeycomb-like structure with multiple vesicles embedded in a single lipid bilayer structure [47,54,55,56]. Liposomes ranging between 50 and 450 nm^3^ in volume are used for medical applications. SUVs are most commonly used in drug-delivery applications due to their uniform encapsulation of drugs and longer circulation times [54,55,56].

Moreover, liposomes are among the most researched NPs partly because of the ease of their surface modification and functionalization to better fit the application as per the tumor pathophysiology. Functionalized liposomes showed significant improvement in physiological behavior when compared to non-functionalized conventional ones **(**Figure 2). To enhance the liposomes’ physical stability in the bloodstream, increase the fluidity of their membranes, and prolong their retention time, natural sterols such as cholesterol are added. Cholesterol is a hydrophobic molecule that interacts with the core of the liposomal membrane and helps reduce its permeability to water [36,42]. This, in turn, increases the liposomal membrane micro-viscosity and fluidity by making them less rigid, prevents crystallization of the phospholipid acyl chains, and increases their stability in the presence of blood/plasma, both in vivo and in vitro. [57,58]. Furthermore, cholesterol can anchor or attach polyethylene glycol (PEG) chains, shielding them from abrupt recognition and elimination from the body and rendering them thermodynamically and sterically stable. PEGylation enhances the half-life of the nanocarrier in the bloodstream, as non-stealth liposomes get rapidly phagocytosed by the reticuloendothelial system (RES) and cleared from the body [42,59,60].

PEG chains are soluble in aqueous and organic solvents, highly biocompatible, easily synthesized, have a linear or branched structure, and show low immunogenicity [61]. The chains can vary in length and configuration, and they are grafted into the liposomes through linkers to create the PEGylated liposomes, commonly referred to as stealth liposomes. Noble et al. [62] claimed that optimum circulation times could be achieved by incorporating five mol% polyethylene glycol with a molecular weight of 2000 g/mol, PEG2000, into the formulation and elaborated the use of PEG2000 is “based more on tradition rather than scientific reasoning.” The choice of the linker is essential, as it alters the extent to which the PEG chains are implanted into the liposomal membranes and can also impose behavioral changes on the liposomes. For example, phosphate linkages are suspected of provoking opsonization, while ester linkages, which are pH-sensitive, are vulnerable to biological decomposition [62].

A study [61] compared the in vivo pharmacokinetic performance of free DOX with PEGylated and non-PEGylated liposomal formulations. Interestingly, the PEGylated liposomal DOX clearance rate decreased by 100-fold (Cl = 0.023 L/h), and its half-life (t1/2 = 83.7 h) was prolonged by eightfold, compared to free DOX (Cl = 25.3 L/h, t1/2 = 10.4 h). Moreover, the distribution volume decreased significantly from 364 L to 139 L to 3.0 L in the free DOX, non-PEGylated, and PEGylated liposomal DOX, respectively. This conclusion demonstrated that PEGylation prevents premature drug release and that most of it remained entrapped without leakage. Another study by Awad et al. [59] investigated the effects of PEGylation on the US-mediated release kinetics from calcein-loaded liposomes. This research concluded that the calcein (model drug) PEGylated liposomes were more sonosensitive and presented significantly enhanced release profiles when exposed to pulsed US at 20 kHz. The release profiles of the PEGylated liposomes were higher than the non-PEGylated ones at all tested power densities. It was reported that the PEGylated liposomes released 57.5% ± 4.5 of their contents, whereas the non-PEGylated ones released only 22.7% ± 1.7 by the end of the third US pulse at 12 W/cm^2^ power density. The conclusions of this study further emphasize the effectiveness of PEGylation in enhancing the overall performance of the SDDS.

### Passive and Active Targeting

Primarily targeting cancer cells follows the natural course by targeting anatomical and physiological feature differences between the normal cells and tumor microenvironment. Controlling the physicochemical factors such as size, charge, and hydrophobicity of the liposomes enables passively targeting cancer microvasculature with larger pores sizes compared to normal capillaries’ cells. The tumor vasculature is characterized by its porous and leaky structuring, allowing liposome passage, permeability, and retention, typically <200 nm [63]. This effect is known as the enhanced permeability and retention (EPR) effect, which forms the basis for passive targeting mechanisms (Figure 3A). Since healthy tissues do not allow liposomes to extravasate through their non-porous and tight junctions, this leads to a differentially higher concentration and accumulation of the drug in tumor cells compared to the rest of the body [64].

Some tumor-related factors to consider when utilizing the EPR effect include, but are not limited to, tumor type and density and its vascular permeability as a function of secretion of permeability factors. As far as the nanocarrier design is concerned, their chemical properties, surface functionalization, and charge and morphologies are all considerably impactful aspects [64]. However, designing SDDSs with complete dependence on passive targeting has significant limitations, such as the possible accumulation of the NPs in the spleen and liver as these organs have fenestrated vasculature and the incapability of the NPs to sufficiently penetrate deep enough through the complex tumoral network due to heterogeneities in structure [65]. Moreover, the EPR effect may result in a slower uptake of nanocarriers and delayed drug pharmacokinetics, where the slow drug release would not allow the drug to reach the desired therapeutic concentration. Moreover, passive targeting is limited to certain solid tumors larger than approximately 4.6 mm, with porosity depending on the type and location of the tumor. Furthermore, non-vascularized sites are questionable when taking advantage of the EPR effect; therefore, facilitating the uptake of nanocarriers by the tumor cells and protecting healthy cells by other means of targeting becomes essential. Active targeting mechanisms allow functionalizing the surface of the nanocarriers to make them more affinized towards cancer cells than healthy ones. It can compromise the inadequacies mentioned above, as it depends on specific receptor-ligand interactions between highly expressed cell-surface receptors on the tumor cells and the engineered drug carrier’s surface. NPs can be functionalized with ligands, or moieties, including proteins, poly-peptide sequences, antibodies, enzymes, vitamins, and carbohydrates [66,67,68,69]. Since tumor cells are known to overexpress receptors that participate in growth and survival pathways, such receptors make promising active targets. To this end, nanocarriers could be conjugated to the natural ligands of these receptors to ensure their accumulation and internalization at the tumor site [70]. Ideally, these receptors would be tumor neoantigens as these are overexpressed on tumor cells compared to healthy cells. Therefore, for this system to work, the ligand in question must have a high affinity to the target receptor to elicit a specific response akin to a lock-and-key mechanism (Figure 3B).

## 3. Acoustic Stimulation by Ultrasound

Upon injection into the patient’s bloodstream, the nanocarriers tend to accumulate at the tumor’s leaky vasculature due to the EPR effect. To unleash the full potential of these drug-loaded vehicles, it is mandated to utilize a triggering mechanism to release the encapsulated drug in a controlled, timely, and efficient manner. Ultrasound (US) has gained considerable attention in research as one of the best drug release mechanisms due to its non-invasiveness, safety record, and relatively low costs. Although it is best known in the medical field for its imaging application, i.e., embryo monitoring and imaging, it has developed to become a means of diagnostics and therapies. The US’s mechanism of action relies on its waves. US waves are longitudinal mechanical sound waves that require a medium for transmitting energy transducers containing piezoelectric crystals that produce acoustic waves as an alternate electrical current is converted into mechanical energy. When an electric pulse is generated and sensed by the crystal, it vibrates; consequently, the surrounding medium experiences pull and push forces, and thus waves are generated [71].

There are two main mechanisms by which US impacts cells and tissues in therapeutic applications, i.e., thermal and mechanical effects [71]. Sonicated tissues experience thermal effects due to hyperthermia, whereby exposed tissues experience an overall increase in the medium’s temperature. The extent to which the medium absorbs energy is a function of multiple factors, such as the frequency of the US and the exposure time. Moreover, some factors are intrinsic to the medium, such as its absorption coefficient; the higher the value of this coefficient, the more thermal effects will be experienced by the tissues [71]. The impact of US on drug release from liposomes is a well-developed area of research [72,73,74]. When the acoustic waves interact with liposomes, some of the acoustic energy will be dissipated and absorbed by the phospholipid bilayer, causing an increase in temperature, which in turn slightly liquifies the microstructure of these nano vehicles and promotes drug release. In drug-delivery applications and to achieve hyperthermia, the tissue temperature should not exceed 43 °C. Within a temperature range of 40 to 43 °C, the increase in temperature accompanied by the rise in blood flow causes vasodilation as well as an increase in the permeability of the tumor’s vasculature, hence enhancing the accumulation of nanoparticles at the diseased site. However, strong hyperthermia, which occurs when temperatures increase beyond 43 °C, could cause necrosis to both healthy and cancerous cells and cause severe burns [73].

The other mechanism by which US induces biological effects is mechanical, mainly through cavitation events. Bubbles pre-exist or are generated in the fluid due to the pressure dropping below the liquid’s vapor pressure. The pressure drop could be induced by exposure to US waves. Acoustic cavitation occurs when these cavitation nuclei—gas-filled bubbles in the insonated liquid media—form, grow, oscillate, and eventually collapse [75]. It has two primary modes; stable and transient [76]. A specific phenomenon of transient cavitation occurs when the bubbles are bounded by any given biological boundary from one side while the other side oscillates freely; it is referred to as asymmetric collapse. The bursting bubbles do not generate a regular shock wave that propagates in spherical dimensions, but rather the energy from the collapse is directed inwards towards the center of the bubble from the free side, propagating linearly. This non-spherical collapse induces high-speed energy-intensive acoustic microjets. The shock waves and shooting microjets can cause neighboring liposomal membranes to burst, thus promoting drug release [77]. Similarly, if the bubbles are close to the tumor site, their collapse can induce the formation of pores in the plasma membranes of the cells in a process called sonoporation [78,79]. This further enhances the accumulation of the drug at the tumor site. When considering the application of US in drug delivery, collapse cavitation has a more predominant role compared to stable cavitation. The former has been shown to enhance drug uptake, and the payload delivery to the individual cells as their permeability is altered with the aid of shock waves and microjets. In contrast, stable cavitation has some effects on changing the overall permeability of diseased vessels.

The application of ultrasound for the treatment of organs such as lungs pose obstacles to acoustic energy due to their respiratory movement, access limitations offered by lung-enclosing ribs, and air-filled spaces that derange the acoustic energy deposition due to impedance mismatch between the ventilated lungs and the adjacent tissues, causing total absorption or total reflection of the acoustic energy [80]. While research ventures to new methods for utilizing US safely, an in vivo investigation tested a lung flooding technique by infusing saline into the lungs prior to applying high-intensity focused ultrasound (HIFU), which aided in matching the impedance while keeping the patient ventilated. This helped eliminate the total absorbance or reflection of acoustic energy, with highly selective penetration of HIFU causing a local increase in the temperature of ex vivo tumor tissue by 7.5-fold than the flooded lung tissue. Furthermore, flooded lung tissues showed a 100% successful sonographic tumor detection rate compared to atelectatic lungs, which were only 43% [80,81].

Clinical studies conducted by Idbaih and colleagues [82,83] on patients with glioblastoma revealed improved penetration and enhanced efficacy of carboplatin chemotherapy through the disrupted blood–brain barrier (BBB) with the help of the implantable low-intensity pulsed ultrasound (LIPU) device SonoCloud-1 (SC1). The device emitted US waves with a resonance frequency of 1.05 MHz, coupled with Sonovue microbubbles (Bracco). Disruption of the BBB was observed to be proportional to an increase in acoustic pressures, with no sign of dose-limiting toxicity [82]. Whereas, Sonocloud-9 (SC9), with a similar concept currently undergoing clinical research, proves the potential of US-mediated BBB disruption and enhancement of chemotherapeutic efficacy.

Aryal et al. [84] investigated the delivery of liposomal doxorubicin (Lipo-DOX) for the treatment of glioma in rats through disrupted BBB induced by focused US (US parameters: 0.69 MHz frequency, 10 ms bursts, pulse repetition frequency 1 Hz, 0.55–0.81 MPa, 60 s) coupled with microbubbles. The study reported the absence of side effects such as neurotoxicity; however, capillary damage was observed during sonication due to inertial cavitation. It was recommended to use lower-pressure amplitude ultrasound to avoid inertial cavitation.

Perfluorocarbon (PFC) gas nanobubbles, ranging in size from 450–690 nm, are used as US contrast agents, as they allow enhancement of image contrast upon vaporization and backscattered echoes. First exploited by Effinger and Wheatley [85], they found that these nanobubbles extravasated through leaky tumor vasculatures and contributed to in vitro image enhancement of breast cancers. Wheatley et al. [86] further investigated ST68 nanobubble-assisted acoustic enhancement in vitro and in vivo and demonstrated nanobubble oscillation upon US irradiation, which resulted in an enhancement of more than 20 dB. Besides enhancing imaging efficiency, nanobubbles can improve the therapeutic performance of nanocarriers when accompanied by US. Wang et al. [87] discovered the significant inhibition of cell growth in human prostate cancer cell lines (C4-2, LNCaP, and PC-3 cells) by employing 609.5 ± 15.6 nm-sized nanobubbles coupled with ultrasound. Upon sonication, nanobubbles encapsulated with AR siRNA showed the lowest expression of AR mRNA in the C4-2 prostate tumor xenograft mouse model compared to the rest of the groups in the study.

To increase liposomal sonosensitivity, microbubbles/nanobubbles or phase shift emulsion droplets that are highly sensitive to ultrasound can be incorporated or coupled with liposomes and can enhance liposomal sonosensitivity and dramatically increase the drug uptake by cancer cells [88]. Several studies have been conducted on these enhancers of ultrasound-mediated drug delivery [89] in targeting renal cell carcinoma [90], prostate cancer cell lines [87], breast cancers [91], and brain cancers [92].

Along the same lines, echogenic liposomes and bubble liposomes increased the sensitivity to ultrasound by sonoporation of liposomes and cancer cell membranes. FDA-approved microbubbles filled with a hydrophobic gas phase, e.g., PFC stabilized by a shell of lipid or polymer, can oscillate and generate contrast in response to US pressure variations. However, the microbubbles’ larger size relative to pores in the tumor’s leaky vasculature did not allow extravasation into cancer cells [93,94,95]. This limitation was, however, overcome by employing multifunctional nanocarriers with perfluorocarbon gas cored or PFC nanodroplets/emulsions that convert into echogenic microbubbles upon heating to physiological temperatures or/and ultrasonic negative peak pressures. The coalescence of microbubbles in tumor cells, followed by cavitation and eventual disintegration, results in the release of the encapsulated drug at the tumor site [96,97,98,99].

Microbubbles can be introduced into the liposomes with other therapeutic compounds to promote cavitation and sonoporation (pore formation by applying acoustic US) and release the drug at the tumor site. A study by Ingram and colleagues showed a significant increase in the efficacy of cytotoxic low-dose medicines, irinotecan, and SN38, by triggering microbubbles using the US in colorectal cancer mouse models [99].

Olsman et al. investigated the effect of focused ultrasound (FUS), and microbubbles on the transferrin (Tf) targeted liposomes in enhancing the permeability of the blood–brain barrier in rats, overexpressing TfR in the BBB. The study revealed that FUS and microbubbles helped safely increase blood–brain barrier permeability and recorded a 40% increase in the accumulation of Tf-targeted liposomes in the brain hemisphere compared with isotype immunoglobulin G (IgG) liposomes. However, the size of microbubbles, i.e., 1µm or above, limits them within the tumor vasculature and prevents microbubbles from penetrating the tumor cells. Thus, they have been used as intra-vascular agents to actively target endothelial markers such as VEGFR2 and αvβ3 integrin. The size restriction of microbubbles introduced the use of nano-scale-sized nanobubbles and nanoemulsions that would easily extravasate into the tumor tissues and get endocytosed into the tumor cells [100,101,102,103].

Emulsions on a nanoscale are called nanoemulsions, and they have nanodroplets of liquid dispersed through another immiscible liquid. Liposomes encapsulate phase shift nano-sized liquid droplets such as perfluorocarbons (PFCs, with a low boiling point) for drug delivery applications. PFCs enhance the sensitivity of liposomes to the US waves. Upon exposure to the US, during the low-pressure wave, the pressure around emulsion droplets falls below the vapor pressure, and they vaporize, resulting in the expansion and explosion of the liposome. Lipid bilayer liposomes can only undergo 3% expansion in their structure before they break or puncture; this aids in releasing the encapsulated drug at the tumor site [35,66].

Perfluorocarbons (PFCs) are considered excellent candidates for emulsions in drug delivery applications due to their biocompatibility, non-toxicity, and hydrophobic behavior; hence, they have a very low solubility in aqueous solutions or even blood. In medicine, PFCs find their application as ultrasound imaging contrast agents and oxygen carriers in blood substitutes [35,66,102,103].

In a study by Lattin et al. [103], the behavior of PFC_5_ emulsion droplets upon exposure to the US demonstrates that only tiny emulsion droplets were visible before the application of US; however, upon exposure to the US, tiny emulsion droplets had vaporized into large gas bubbles. Gracia et al. [104] investigated the phase-transitioning behavior of perfluorocarbon nanodroplets to microbubbles under US negative peak pressure and their application as ultrasound contrast agents for in vitro human breast carcinoma-derived cell line SK-BR-3, confirming their stability with significant image enhancement in B mode of US.

## 4. Ultrasound-Activated Agents as Nanocarriers

eLiposomes are liposomes with encapsulated nanoemulsions that help in the rupturing of the drug-carrying liposomal membrane upon sonication, leading to the vaporization of PFC liquid. The bursting of the liposome membrane causes a faster release of the encapsulated drug compared to conventional liposomes, as liposomes sustain only a 3% expansion in volume and consequently rupture due to expansion [105]. Nanoemulsions are nano-sized emulsions with two immiscible phases, a hydrophilic phase and a hydrophobic phase, that are thermodynamically unstable. Protein and lipids are used as surfactants to stabilize nanoemulsions [106]. PFCs are commonly used as nanoemulsions, because of their hydrophobic nature, biocompatibility, and non-toxic, and stable organic compounds. PFC droplets are non-carcinogenic fluoroalkanes extensively used as contrast enhancers in ultrasound imaging. These nanoemulsions undergo phase shifts and convert into echogenic microbubbles, making them good candidates for ultrasonic imaging or nuclei for cavitation events necessary for effective smart drug delivery. PFCs’ high interfacial energy in water makes them an attractive choice; perfluoropentane (PFP), in particular, has been extensively used due to its low boiling point close to physiological temperatures of 29 °C and requires low acoustic amplitude to induce vaporization [105,106,107,108,109,110]. Koroleva and Plotniece [111] studied the stability of nanoemulsions within liposomes using Langevin dynamics. It was reported that the stability of nanoemulsions with higher fractions of dispersed phase highly depends on their ζ-potentials. It was found to be higher than 40 mV, in addition to a surfactant layer, to reduce the toxicity of dispersed phase droplets.

The phase transition of PFC droplets from liquid to gas under the effect of an acoustic wave is called acoustic droplet vaporization (ADV) (Figure 4). This occurs when the gas’s vapor pressure is no longer in equilibrium with its liquid state at a specified temperature, and the liquid quickly escapes to the gaseous phase [112]. Furthermore, Laplace pressure is the pressure imposed on the interior of the droplet because of the interfacial energy of surface tension between the two immiscible phases compressed within the droplet. This phenomenon is widely being used to increase the efficacy of drug delivery applications.

Gao et al. enhanced the ultrasonic sensitivity of DOX micelles by incorporating perfluoropentane (PFP) emulsion nanodroplets. Nanodroplets with block copolymers poly(ethylene oxide)-block-poly(L-lactide) (PEG-PLLA) and poly(ethylene oxide)-block-poly(caprolactone) (PEG-PCL) vaporized into nanobubbles upon insonation by acoustic droplet vaporization [96,112,113], which passively accumulated at the tumor site and demonstrated a 15% decrease in size from 770 ± 86 nm to 674 ± 72 nm after heating for 4 h at 37 °C. Nanobubbles coalesced into microbubbles and synergistically enhanced the ultrasound-mediated DOX delivery in ovarian cancer A2780 cells in vitro. For in vivo studies of breast cancer, MDA MB231 in nu/nu mice reveals a regression of tumor size post administration of drug-loaded microbubbles and sonication for 150 s, with 2 W/cm^2^ at 3 MHz with a duty-cycle of 20%. However, in the absence of ultrasound, drug release did not occur and was retained [97]. This was confirmed by Rapoport et al., where non-thermal pulsed ultrasound (20% duty cycle) was used to vaporize PFP and (perfluoro-15-crown-ether) PFCE nanodroplets to larger-sized microbubbles, as a result of the coalescence of nanobubbles, which can further enhance echogenicity [114].

Several studies have reported the acoustically stimulated response of eLiposomes due to the acoustically induced thermal and mechanical effects. Upon exposure to a lower ultrasound pressure wave, nanoemulsions encapsulated inside liposomes vaporize, resulting in an increased radius by fivefold and reduced interfacial tension 3 dyn/cm (for DPPC). This causes the liposomes to expand and burst, releasing the drug at short exposure times. Blasting of liposomes produces shockwaves or liquid jets that induce sonoporation of the cell membrane, further enhancing membrane permeability [66,72,115,116]

Lattin et al. [105] were the first to utilize the novel technique of encapsulating monodisperse phase-shifting perfluorocarbons (PFC_6_) nanoemulsion droplets within the liposomes to enhance liposomal sensitivity to ultrasound. PFC emulsion droplets were reported to undergo acoustic droplet vaporization within the liposomal core, a phenomenon by which liquid emulsion vaporizes to gas bubbles and cavitates upon exposure to the low-pressure phase of ultrasound. Upon endocytosis and US exposure, expansion of PFC gas helps rupture the endosomal membrane and release the drug directly into the cytosol of the cell, thus, as discussed earlier, increasing ultrasound responsiveness and overcoming size limitations posed by microbubbles. The authors discussed the properties of eLiposomes which highly depend on the type of lipid and emulsion droplet employed. Experimental research has suggested that using PFC_5_ emulsion droplets with a lower boiling point (29 °C) and high vapor pressure can allow controlled vaporization and efficient drug release at physiological temperatures.

Subsequently, they studied the release of a model drug, i.e., calcein, from eLiposomes and explored the role of PFC emulsion droplets in drug release [117]. PFC emulsions interacted with ultrasound and demonstrated sonosensitive behavior by undergoing a phase transition from liquid to large gas bubbles up to 20 µm upon exposure to ultrasound. Lattin et al. varied ultrasound power density and exposure times to demonstrate the release of calcein from eLiposomes with different emulsions droplet sizes and control liposomes with large or small emulsion droplets to their exterior. PFC_5_/PFC_6_ eLiposomes with large and small emulsion droplets showed a significant increase of calcein release of up to three to fourfold compared to control liposomes. The ultrasound exposure time of up to 10 s demonstrated efficient calcein release (94% and 47%) from large/small-PFC_5_ eLiposomes, respectively, compared to control liposomes. This was attributed to rupturing of the liposomal membrane as a result of emulsion vaporization and sonoporation by ultrasound. Calcein release was also proportional to ultrasound power densities, with control liposomes showing the lowest release at every power density compared to eLiposomes. It should also be noted that there was no significant difference in calcein release from control liposomes with and without exterior emulsion droplets.

In another study, Javadi et al. [116] used the technique carried out by Lattin et al. [117] to enhance drug delivery into HeLa cells by encapsulating perfluorocarbon emulsion droplets (perfluoropentane PFC_5_/perfluorohexane PFC_6_) within the liposomal core. This study used low-frequency (1 W/cm^2^ power density and 20 kHz frequency) ultrasound to irradiate folated emulsion liposomes in HeLa cells for 2 s. The researchers also demonstrated the phase transition of PFC emulsion droplets from the liquid to gaseous phase with liquid-to-gas expansion ratios of 137-fold and 125.9-fold PFC_5_ and PFC_6_. Sonication of folate-conjugated eLiposomes demonstrated an improved in vitro uptake calcein delivery in HeLa cells compared to non-targeted eLiposomes or targeted conventional liposomes without PFC_5_ emulsions. It should be noted that targeting ligands on the surface and phase-transitioning emulsions in the core had a synergistic effect on calcein delivery to HeLa cells. eLiposomes were found to be ultrasound responsive to low power densities and short exposure duration and overcome the size limitation posed by microbubbles in helping extravasation through tumor vasculature.

In a following study, the researchers [118] further investigated the delivery of calcein and plasmid transfection into HeLa cells using low-frequency ultrasound (1 W/cm^2^ power density and 20 kHz frequency) to trigger eLiposomal vehicles with folate conjugated to their surfaces. eLiposomes were prepared using the same method from the previous study [116]. Javadi et al. conducted parallel experiments to study the role of emulsion droplets, targeting ligands, and ultrasound for drug and plasmid transfection under identical optical conditions. By comparing relative fluorescence, it was found that non-insonated eLiposomes showed negligible calcein release; on the other hand, insonated eLiposomes showed significantly high calcein release. In this study, the authors demonstrated a more efficient calcein release and plasmid transfection as folated eLiposomes were endocytosed and released their cargo into the cytosol of cells upon exposure to ultrasound. In contrast, negligible relative fluorescence was observed in the absence of emulsions or targeting ligands. The authors also highlighted the stability of eLiposomes in retaining uncoated plasmids without degradation post-endocytosis.

The in vitro release of doxorubicin from folated PFC_5_ eLipoDox (eLiposomes sequestering Dox) in HeLa cells upon exposure to low-intensity ultrasound was also investigated [119]. Exposure to ultrasound (1 W/cm^2^, 20 kHz) at increasing exposure times from 2–60 s released 80% of Dox from eLipoDox and only 50% from LipoDox. This was attributed to the vaporization of PFC_5_ nanoemulsions within the liposomal core. Dox release further increased upon increasing power intensities; however, Dox release from both eLipoDox and LipoDox was observed to decrease as frequencies were increased. Furthermore, it was demonstrated that folate aided the uptake of eLipoDox by HeLa cells and raised the fluorescence intensity sixfold when compared with the release from eLipoDox with blocked folate receptors. The authors hypothesized that the disruption of liposomal and endosomal membranes, as a result of acoustic droplet vaporization, was responsible for the Dox release directly into the cytosol and significantly increased cytotoxicity compared to free Dox. Thus, eLiposomes sequestered Dox, decreasing systemic toxicity and minimizing Dox-induced side effects such as cardiotoxicity.

Along the same lines, Husseini et al. [120] investigated the behavior and stability of eLiposomes at elevated temperatures by monitoring the release of calcein after incubating for 15 min in order to confirm the premature vaporization of PFC_5_ in the absence of ultrasound. It was suggested that heterogenous nucleation might be the possible mechanism responsible for the disintegration and release at high temperatures due to dissolved oxygen/nitrogen in PFC_5_, the presence of contaminants in the emulsion droplets, or other sources. Results showed that release increased exponentially as a function of temperature; however, eLiposomes remained stable at physiological temperatures (37 °C), which was above the boiling point of PFC_5_ (29 °C). This was attributed to the Laplace pressure imposed on the PFC_5_ droplets, which raises the boiling point, keeping them stable well above their boiling point. eLiposomes do not show premature breakage or release content until exposed to ultrasound, thus, potentially facilitating temporal and spatial control over drug release in drug delivery systems.

In another study, eLiposomes were tested in vitro to investigate their uptake and release of high-molecular-weight cytotoxic mistletoe lectin-1 (ML1) and protein horseradish peroxidase (HRP) when exposed to HIFU. Parallel experiments were conducted to study the role of PFC nanoemulsions in drug release [121]. It was demonstrated that eLiposomes emitted stronger signals at second harmonic, subharmonic, and broadband noise signals than PFC nanoemulsions and conventional eLiposomes. eLiposome signals further increased upon increasing peak negative pressure from 1.5 MPa to 3.0 MPa, thus helping in forming cavitating bubbles inside the liposomes and confirming the importance of nanoemulsions in the interior of liposomes. eLiposome with intermediate DSPE-PEG2000 amine was reported to be highly stable and released the highest ML1 upon exposure to 24 MPa HIFU for 1 min inhibiting CT26 cell growth. US thermal effects were reported to be negligible. Table 2 provides a summary of the different US-enhancing agents used in medical applications, while Table 3 presents a summary of some in vitro reports about the response of acoustically-stimulated eLiposomes due to induced thermal and mechanical effects.

## 5. Concluding Remarks

Due to the detriment to the patient’s quality of life and the potential lethality of some of the side effects associated with the common treatments, researchers and professionals are always venturing to find alternative novel therapeutic modalities to preserve the quality of cancer patients’ lives. Such ventures aim at reducing the adverse side effects associated with the available therapies, as well as to have more targeted and efficient treatments. One of the most well-established advances is state-of-the-art nanocarriers incorporated in SDDSs. SDDSs are nanoplatforms with important characteristics and bio-functions that make them optimal for the remote delivery of drugs to targeted sites under controlled release conditions. Also, the release mechanisms of such systems are controlled and can be tuned to be stimuli-responsive to endogenous or exogenous triggers, which is an added controlling advantage. Disadvantages associated with traditional chemotherapy, such as nonselective systemic activity, poor drug solubility, hepatic biodegradation, dose-limiting toxicity, and the deterioration of healthy cells, can all be overcome using nanoparticles (NPs) as drug delivery vehicles.

Conclusively, ultrasound-mediated eLiposomal drug delivery has the potential to significantly improve the therapeutic effects of chemotherapeutic drug delivery. With comparatively faster drug release at short insonation times, it is a complex process that requires acoustic activation of phase-shifting PFC nanoemulsion droplets to vapor, which shear opens the nanocarriers and allows the release of the drug to the cells. eLiposomal application still holds scope for improvement in order to fully exploit the benefits of ultrasound parameters for drug-delivery applications. Further research must be conducted on eLiposomal design considerations, developing uniform-size phase-shift PFC nanoemulsion droplets and PFC emulsion droplet incorporation in combination with other PFCs. Furthermore, PFC nanoemulsions droplets and their use in drug delivery require in vivo proof with a focus on eLiposomal interaction with ultrasound in an in vivo mimicking environment and associated bioeffects with acoustic droplet vaporization.

## Figures and Tables

**Figure 1 pharmaceutics-15-00421-f001:**
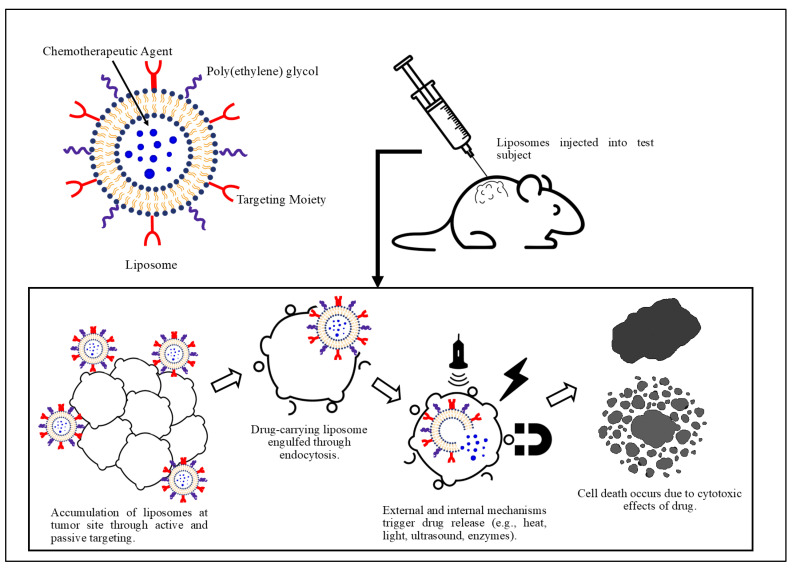
Schematic illustration of liposome-based drug delivery system for the treatment of solid tumors.

**Figure 2 pharmaceutics-15-00421-f002:**
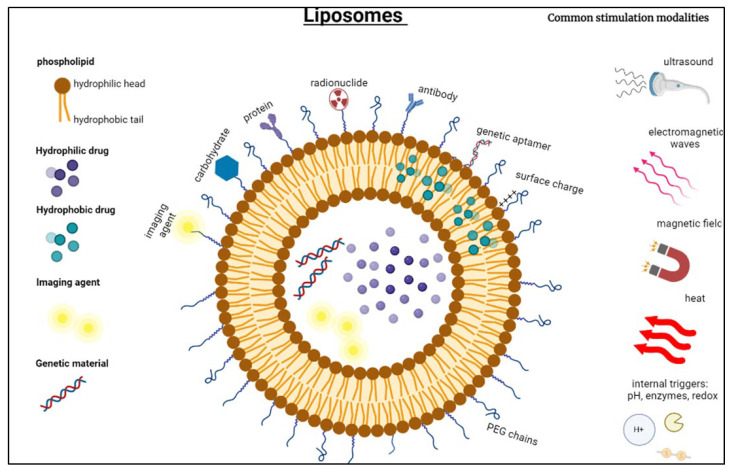
A schematic illustration of liposomes, their functionalization, and common triggering modalities to release the encapsulated payload from liposomes.

**Figure 3 pharmaceutics-15-00421-f003:**
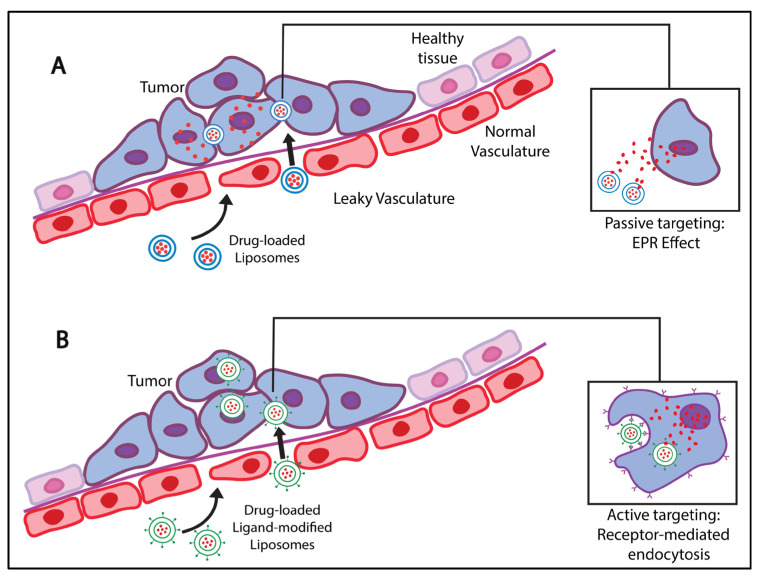
(**A**) Passive targeting. Liposomes accumulate and retain at the target site with the help of leaky vasculature and a defective lymphatic drainage system (EPR effect); (**B**) Active targeting. Liposomes accumulate via a passive targeting mechanism; drug carriers are endocytosed by specific receptor-mediated interactions between the overly-expressed tumor cell-surface receptors and targeting ligands on the liposomes’ surface.

**Figure 4 pharmaceutics-15-00421-f004:**
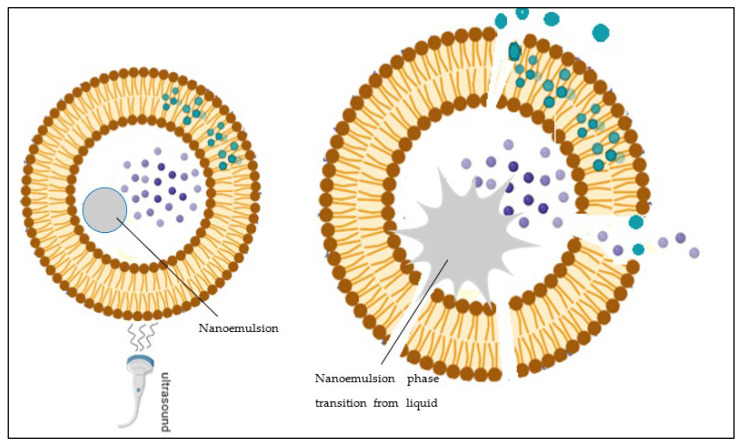
Schematic representation of ADV and bursting of liposomes upon sonication.

**Table 1 pharmaceutics-15-00421-t001:** Summary of the liposomal drugs approved by the FDA and/or EMA for cancer treatment [28,29,30,31,32,33,34,35,36,37,38,39,40].

Product™	Encapsulated Drug, Administration Route	Approved Year/Area	Indication	Composition	Size
Doxil	Doxorubicin, IV	1995, FDA	Ovarian, breast cancer, Kaposi’s sarcoma	HSPC, PEG-DSPE, chol	SUVs (100 nm)
Caelyx	Doxorubicin, IV	1996, EMA	Ovarian, breast cancer, Kaposi’s sarcoma	HSPC, PEG-DSPE, chol	SUVs (100 nm)
DaunoXome	Daunorubicin, IV	1996, FDA	Kaposi’s sarcoma	HSPC, DSPC, Chol	SUVs (45–80 nm)
Myocet	Mifamurtide, IV	2000, EMA	Metastatic breast cancer	EPC, Chol	MLVs (80–90 nm)
Mepact	Mifamurtide PE	2009, EMA	Osteosarcoma	POPC, OOPS	MLVs (2.0–3.5 μm)
Marqibo	Vincristine, IV	2012, FDA	Acute lymphoid leukemia	SM, Chol	SUVs (130–150 nm)
Lipusu	Paclitaxel	2013, FDA	Gastric, ovarian, and lung cancer	Non-modified liposomes	400 nm
Onyvide	Irinotecan, IV	2015, FDA 2016, EMA	Metastatic adenocarcinoma of the pancreas	DSPC, MPEG2000-DSPE, Chol	SUVs 110 nm
Vyxeos	Daunorubicin and cytarabine	2017, FDA 2018, EMA	Acute myeloid leukemia	DSPC, DSPG, Chol	110 nm
Zolsketil	Adriamycin, IV	2022, EMA	Metastatic breast and ovarian cancer, multiple myeloma, and Kaposi’s sarcoma	HSPC, PEG-DSPE, chol	SUVs (100 nm)

Abbreviations: IV intravenous; FDA Food and Drug Administration; EMA European Medicines Agency; HSPC hydrogenated soy phosphatidylcholine; PEG polyethylene glycol; DSPE 1,2-distearoyl-sn-glycero-3-phosphoethanolamine; SUVs small unilamellar vesicles; MLVs multilamellar vesicles.

**Table 2 pharmaceutics-15-00421-t002:** Comparison between different US-enhancing agents [87,97,106,115].

	Microbubbles	Nanobubbles	Nanoemulsions
Size	1–10 µm which limits accumulation to the tumor vasculature (380–780 nm)	200–300 nm to pass through the tumor vasculature and destruct upon ultrasound irradiation	10–1000 nm to improve their stability and vaporize to form large microbubbles upon US irradiation
Circulation stability	Short circulation time (a couple of minutes)	longer circulation time	due to low solubility, PFC gases remain stable for much longer in aqueous solutions in comparison with air bubbles
Physical structure	Micron-sized gas core stabilized by polymer, lipid, or protein surfactants with low drug loading capacity	Sub-micron-sized gas core stabilized by polymer, protein, or lipid surfactants with high drug loading capacity	Same lipid or different lipid layers can be used as a surfactant for nanodroplets as well as the liposomes encapsulating them with high drug-loading capacity
Echogenicity	Excellent echogenicity and enhancement of membrane permeability by sonoporation	Echogenicity is smaller compared to micron-sized bubbles	limited echogenicity compared to microbubbles
Action mechanism	Upon ultrasound irradiation, micro-scaled microbubbles may collapse and release the drug outside the tumor cells, leading to a decreased anticancer efficacy	Upon ultrasound irradiation, nanobubbles cavitate, collapse, and release the drug within the tumor cells.	Upon local ultrasonic irradiation, nanoemulsion droplets vaporize into microbubbles and enhance the intracellular drug uptake by tumor cells, providing a spatial control of up to a few millimeters or sub-millimeters.

**Table 3 pharmaceutics-15-00421-t003:** A summary of some in vitro studies on eLiposomes triggered by US.

eLiposome Composition	Targeting Ligand/Targeted Cancer Cells	US Parameters	Load	Remarks	**Ref.**
DPPA/PFC_5_ emulsion droplets (100 nm) encapsulated by DMPC, DSPE-PEG2000-amine liposomes (200 nm)	Folate/HeLa cancer cells	varying power densities (0.25–1 W/cm^2^) and variable exposure for 2–6.4 s	Calcein and plasmid protein	Low-intensity US insonated eLiposomes showed significantly higher uptake of calcein or plasmid transfection by HeLa cells relative to non-insonated eLiposomes. The presence of encapsulated emulsion droplets and folate moiety had a synergistic effect in acoustically-stimulated drug delivery within the cell’s interior.	[118]
DMPC/Cholesterol/DSPE-PEG2000-amine liposomes (200 nm) encapsulated with DPPC/PFC_5_ nanoemulsions (100 nm)	Non-modified/murine CT26 colon carcinoma cells	High-intensity focused ultrasound (peak negative pressure 2–24 MPa, frequency 1.3 MHz)	Mistletoe lectin-1 (ML1) and protein horseradish peroxidase (HRP)	It was demonstrated that eLiposomes were emitted stronger at second harmonic, subharmonic, and broadband noise signals compared to PFC nanoemulsions and conventional eLiposomes further increased with the increase in ultrasound peak negative pressures. HIFU-exposed liposomes demonstrated fourfold more cytotoxin uptake and apoptosis than the ones not exposed to HIFU. Ultrasound-sensitive liposomal formulations with less DSPE-PEG showed the highest stability and ultrasound sensitivity with the highest drug release of 80%. Studies reported negligible thermal effects with inhibition of tumor cell viability primarily induced by cavitation of echogenic PFC emulsion droplets.	[72]
eLipoDox-DPPC/Cholesterol/DSPE-PEG2000-amine liposomes encapsulated with DPPC/PFC_5_ nanoemulsions	Folate/HeLa cancer cells	1 W/cm^2^ power density, 20 kHz frequency, and 100% duty cycle for 2 s	Doxorubicin	The cytotoxicity of acoustically-stimulated folated eLipoDox is sixfold higher than non-folated eLipoDox or free Dox. Acoustic stimulation helps release 80% of the Dox from folated eLipoDox, whereas LipoDox without encapsulated emulsions only released 50% of the encapsulated Dox.	[119]
DSPE-PEG2000 amine/DMPC liposomes (200 nm) encapsulated with DMPC/PFC_5_ nanoemulsions	Folate/HeLa cancer cells	1 W/cm^2^ power density, 20 kHz frequency for 2 s	Calcein	Ultrasound-mediated disruption of liposomes with folated surfaces and emulsions encapsulated within them showed significant calcein delivery into HeLa cells compared to conventional liposomes or non-targeted liposomes. Results demonstrate the necessity of both sonosensitive emulsions and targeting ligand (folate) to enhance drug delivery.	[116]
DMPA/DPPC/Cholesterol/DSPE-PEG2000-amine liposomes encapsulated with DPPA/PFC_5_ nanoemulsions	Avidin/hemagglutinating virus of Japan (HVJ)/MCF-7 Human breast cancer cells	1.2 W/cm^2^ power density, 1 MHz frequency, and 30% duty cycle for 30 s	Calcein and phenylphenanthridinium diiodide (PI) fluorescent dyes	Avidin/HVJ co-modified ligands helped bind PFC_5_ emulsion liposomes to the MCF-7 cells, whereas ultrasound irradiation caused acoustic droplet vaporization of PCF5 and lowered the cell viability of MCF-7 cells by 43%. Ultrasound induces cavitation-aided shear force/mild hyperthermia that can temporarily enhance the permeability of tumor vasculature and aid in liposomal endocytosis and drug uptake	[121]

## Data Availability

Not applicable.

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
