# Peer review of "Acoustically-Activated Liposomal Nanocarriers to Mitigate the Side Effects of Conventional Chemotherapy with a Focus on Emulsion-Liposomes"

_pharmaceutics, 2023, doi:10.3390/pharmaceutics15020421_

Round 1

Reviewer 1 Report

I have gone through the manuscript and have some suggestions to improve the quality of the article. This review summarised the recent advancement of liposomal nanocarriers and their release profile under ultrasound for cancer therapy. It will be great if the authors could include some more representative figures for a better understanding of the concept.

Reviewer 2 Report

Dear editors and authors:

The authors summarized the acoustic stimulation of different sonosensitive liposomal formulations in the manuscript. Especially, the advantage of emulsion liposomes was addressed.  But the part or content which liposomal formulations are less toxic than drugs alone were missing, and the manuscript title didn’t consistent with the main text and the abstract. Based on the criteria of the Journal of pharmaceutics, I decided to Reconsider after major revision.

Major comments:

1.      The part or content which liposomal formulations are less toxic than drugs alone were missing, and the manuscript title didn’t consistent with the main text and the abstract. What is the conclusion of the manuscript?

2.      As we know, some disadvantages of the US need to be pointed out in the manuscript. Firstly, cavitation skin burns due to the presence of air between the transducer and the body surface. This also limits the treatment of extensive superficial regions, such as breast cancer, and regions where the air is inherently present, such as lungs and intestines. The other challenge is to focus on organs that have movement.

3.      What are the authors want to say in table 2?  The title of table 2 was hard to understand and didn’t mention in the content. “Table 2. Comparison of different liposomal acoustic stimulation enhancers:[81], [91], [99], [108]”

4.      I didn’t know what is the meaning of the paragraphs?  The authors listed the content of the reference one by one, why?

e.g. “Lattin et al. [111]……..

“Lin et al. [113] ……….”

“Husseini et al. [114] ……..”

Reviewer 3 Report

The topic of the review is interesting but in my opinion treated in a non-exhaustive way. acronyms are not specified (eg HFU) tables are difficult to read. Also missing is a description of what we are what the authors call eliposomes. I believe that the review should be reorganized by better specifying how the systems they talk about are obtained.

Round 2

Reviewer 2 Report

Dear editors and authors:

I’m very glad the authors improved the manuscript, I suggested accepting the manuscript after minor revision.

Minor :

  The title was too general, I suggested the authors revise it to “Acoustically stimulated eliposomes for cancer therapy: a review.” Because the authors mainly talked about the eliposomes.

Author Response

Dear reviewer.,

We have changed the title of the paper to read:

"Acoustically-activated Liposomal Nanocarriers to Mitigate the Side Effects of Conventional Chemotherapy with a Focus on Emulsion-Liposomes"

Best wishes,

Ghaleb